# The Neuroprotective Effect of Erythropoietin on the Optic Nerve and Spinal Cord in Rats with Experimental Autoimmune Encephalomyelitis through the Activation of the Extracellular Signal-Regulated Kinase 1/2 Signaling Pathway

**DOI:** 10.3390/ijms25179476

**Published:** 2024-08-31

**Authors:** Gloria Aleida Pérez-Carranza, Juliana Marisol Godínez-Rubí, María Guadalupe Márquez-Rosales, Mario Eduardo Flores-Soto, Oscar Kurt Bitzer-Quintero, Ana Cristina Ramírez-Anguiano, Luis Javier Ramírez-Jirano

**Affiliations:** 1Centro Universitario de Ciencias Exactas e Ingeniería, Universidad de Guadalajara, Guadalajara 44840, Jalisco, Mexico; gloria.perez5249@alumnos.udg.mx (G.A.P.-C.); ana.ranguiano@academicos.udg.mx (A.C.R.-A.); 2Departamento de Morfología, Centro Universitario de Ciencias de la Salud, Universidad de Guadalajara, Guadalajara 44340, Jalisco, Mexico; marisol.godinez@cucs.udg.mx; 3Laboratorio de Patología Diagnóstica e Inmunohistoquímica, Centro de Investigación y Diagnóstico en Patología, Centro Universitario de Ciencias de la Salud, Universidad de Guadalajara, Guadalajara 44340, Jalisco, Mexico; 4Centro de Investigación Biomédica de Occidente, Instituto Mexicano del Seguro Social, Guadalajara 44340, Jalisco, Mexico; qfb_lupita@hormail.com (M.G.M.-R.); mario.flore@imss.gob.mx (M.E.F.-S.); oscar.bitzer@imss.gob.mx (O.K.B.-Q.)

**Keywords:** erythropoietin, EAE, ERK 1/2 pathway, neuroprotection, optic nerve, spinal cord

## Abstract

Experimental autoimmune encephalomyelitis is a demyelinating disease that causes paralysis in laboratory rats. This condition lacks treatment that reverses damage to the myelin sheaths of neuronal cells. Therefore, in this study, treatment with EPO as a neuroprotective effect was established to evaluate the ERK 1/2 signaling pathway and its participation in the EAE model. EPO was administered in 5000 U/Kg Sprague Dawley rats. U0126 was used as an inhibitor of the ERK 1/2 pathway to demonstrate the possible activation of this pathway in the model. Spinal cord and optic nerve tissues were evaluated using staining techniques such as H&E and the Luxol Fast Blue myelin-specific technique, as well as immunohistochemistry of the ERK 1/2 protein. The EPO-treated groups showed a decrease in cellular sampling in the spinal cord tissues but mainly in the optic nerve, as well as an increase in the expression of the ERK 1/2 protein in both tissues. The findings of this study suggest that EPO treatment reduces cellular death in EAE-induced rats by regulating the ERK pathway.

## 1. Introduction

Experimental autoimmune encephalomyelitis (EAE), a model of multiple sclerosis (MS), associated with local inflammation mediated by T helper cells [1], causes progressive demyelination in the central nervous system (CNS), particularly in the spinal cord [2,3]. In this sense, several experimental pieces of evidence have shown that the migration of T cells to the CNS, the recruitment and activation of glial cells (astrocytes/microglia), and the infiltration of peripheral macrophages are orchestrators of the inflammatory response in EAE. The mediators of this inflammation are tumor necrosis factor α (TNF-α) and nuclear factor kappa B (NFκB), which initiate or amplify MS/EAE [4,5,6]. Furthermore, it has also been shown that an increase in the production of reactive oxygen species and a decrease in antioxidant capacity can lead to demyelination [3,7]. Zargari et al. [8] also demonstrated a reduction in total antioxidant capacity (TAC) in EAE [9].

In this sense, it has been shown that erythropoietin (EPO), through the activation of its specific receptor, exerts neuroprotective effects by inhibiting proinflammatory processes and inhibiting the release of reactive oxygen species (oxidative stress) in re-induced models of Parkinsonism [10], Alzheimer’s disease [11], traumatic brain injury [12], cerebral ischemia [13], etc. Furthermore, EPO prevents neuronal death in the spinal cord in EAE models [14], and this molecule is produced in spinal cord cells such as astrocytes and neurons. It can act as an anti-apoptotic, anti-inflammatory, antioxidant, and trophic factor [15]. However, the mechanism by which EPO exerts its neuroprotective effects through the modulation of inflammatory responses and inhibition of oxidative stress remains unclear, so the present work evaluated the neuroprotective effect of EPO on the optic nerve and spinal cord in an EAE model and whether the extracellular signal-regulated kinase 1/2 (ERK 1/2) signaling pathway mediates this effect.

## 2. Results

### 2.1. Histology

#### Hematoxylin and Eosin/Luxol Fast Blue

In the control group, with hematoxylin–eosin (H&E) staining, the stained myelin sheath’s homogeneity was evident in the tissue, with a clear difference between gray matter and white matter, motor neurons with integrity and without alterations, and an abundance of cells without apparent alterations (Figure 1A–D). 

In the EAE group, with H&E staining, no essential alterations were observed in the cellular structures or the tissue in general (Figure 1E–H). 

No apparent alterations were observed in the EAE+EPO group with H&E staining. The tissue was homogeneous, and the cells present in this tissue were intact (Figure 1I–L). Meanwhile, in the optic nerve, an increase in cellularity is shown in the EAE+EPO group compared to the EAE group, with a value of *p* = 0.053 (Figure 1Q). On the other hand, no apparent alterations were observed in the EAE+EPO+U0126 group with H&E staining, and the tissue was intact and homogeneous (Figure 1M,O). 

With Luxol Fast Blue (LFB) staining in the control groups, the integrity of the tissue is observed without alterations, with intact neuronal projections (Figure 2A–D). 

With LFB staining, EAE animals show signs of demyelination in both the optic nerve and spinal cord, indicated by white areas in the tissue, with a considerable area of loss of myelin integrity; this loss is observed in the optic nerve in the terminal region of the tissue, obtaining a value of *p* = 0.0007 (Figure 2Q), compared to the control group (Figure 2E,F,Q). Something similar occurs in the spinal cord, where an area of demyelination is observed in the dorsal horns of the gray matter (Figure 2G,H).

The optic nerve of the EAE+EPO group appears intact in most of the tissue; in the lower part, a minimal loss of tissue integrity can be observed (Figure 2I,J), obtaining a value of *p* = 0.003, compared with the EAE group (Figure 2R). In the spinal cord, an area of loss of homogeneity is observed in the dorsal horns (Figure 2K,L).

On the other hand, in the spinal cord of the EAE+EPO+U0126 group, an area with loss of myelin integrity is observed in the dorsal horns of the tissue (Figure 2O,P). With LFB staining in the optic nerve, no areas of demyelination are observed in the tissue in this group, and no evidence of injuries caused by the EAE model is shown. Furthermore, there is an increase in the LFB staining intensity in the optic nerve, compared to the EAE+EPO group, obtaining a value of *p* = 0.0018 (Figure 2M,N,Q).

### 2.2. Immunohistochemistry

#### ERK Protein Expression

The IHC of the ERK 1/2 protein in the optic nerve of the control group showed a high protein expression (brown cells) (Figure 3A,E). In the EAE group, a significant decrease in ERK 1/2 expression can be observed compared to the control group (*p* = 0.0013) (Figure 3B,F). In the EAE+EPO group, an increased expression of ERK 1/2 was observed compared with the EAE group, obtaining a value of *p* = 0.0014 (Figure 3C,G,I). The EAE+EPO+U0126 group decreased the expression of ERK protein compared with the EAE+EPO group (Figure 3D,H).

A high protein expression is observed in the IHC of the ERK 1/2 protein of the spinal cord in the control group (Figure 4A,E). The protein expression in the EAE group is lower than in the control groups, with a value of *p* = 0.0008 (Figure 4B,F,I). In the EAE+EPO group, it is observed that the expression of ERK increases in the tissue compared with the EAE group, obtaining a value of *p* = 0.0210 (Figure 4C,G,I), while in the EAE+EPO+U0126 group, a decrease in the expression of the ERK protein was observed because of the U0126 blocker, compared with the EAE+EPO group, obtaining a value of *p* = 0.0026 (Figure 4D,H,I).

## 3. Discussion

In EAE, neurodegenerative lesions of the optic nerve called optic neuritis are part of the pathological process of the CNS, including damage to the brain and spinal cord [16]. EPO is an effective neuroprotective treatment against these lesions based on its expression in the spinal cord and astrocytes. It also promotes the differentiation of oligodendrocytes and preserves the integrity of the myelin sheath [17,18,19]. Likewise, the erythropoietin receptor (EPOr) has been identified in neuronal cells as responsible for neuroprotective effects [17,20]. One of the proposed mechanisms by which EPO exerts its neuroprotective role is the ERK 1/2 pathway, since when entering the nucleus and being activated, it regulates the activity of transcription factors and the gene expression [21]. ERK has been shown to play an essential role in regulating cell proliferation, differentiation, and cell survival. It is also involved in the immune system and autoimmune responses [22] and is crucial in regulating oligodendrocyte development, myelination formation, and myelin sheath thickness [23].

In our model, a decrease in the expression of the phosphorylated ERK protein was observed in the spinal cord of the EAE group, and an increase in the expression of phosphorylated ERK protein was observed in the EAE+EPO group. The protective function of EPO occurs through the EPOr-mediated extracellular signal-regulated kinase (ERK) pathway [19]. EPO treatment enhanced the expression of ERK 1/2 protein in the optic nerve and spinal cord of the EAE+EPO group, and the number of cells increased mainly in the optic nerve of the rats with EAE. The increase in the expression of the ERK 1/2 pathway could induce the activation of the ERK pathway, potentially inducing apoptosis through the positive regulation of p-ERK 1/2, which could activate anti-apoptotic proteins. These results agree with what is reported in the literature, in which it is observed that excessive expression of upstream proteins and kinases in the ERK pathway may be involved in the pathogenesis of neurodegeneration [24].

The expression of the ERK 1/2 in the EAE group decreases the protein expression in the tissue. On the other hand, in the EAE+EPO group, an increase in ERK expression is observed compared to the EAE group. A study by Kilic et al. demonstrated that in an optic nerve transection model, EPO’s protective function in retinal ganglion cells is predominantly driven by the ERK 1/2 pathway [25]. There is evidence that EPO use in neuronal cells can exert a neuroprotective effect through the ERK 1/2 pathway.

Regarding the results shown in H&E staining, in the EAE group, a decrease in the number of cells present is observed in the EAE+EPO group; the effect caused by EAE is reversed. In the group in which U0126 was applied, inhibitor mitogen-activated protein kinase (MEK), an upstream activator of ERK 1/2, was inhibited, simultaneously blocking the ERK and mTOR pathways, thus showing a decrease in the expression of the protein in the spinal cord and optic nerve tissue of the EAE+EPO+U0126 group; this result demonstrates the participation of the ERK 1/2 pathway in the neuroprotection of cell death in the EAE group. 

EPO is currently considered a good candidate for use with a neuroprotective approach in various diseases, such as EAE. The study by Moransard et al. shows us that the main beneficial effect of EPO in EAE resides in the CNS, protecting neurons during the progression of EAE [14]. Neuroprotection in the optic nerve could only be observed as the absence of optic neuritis in the tissue. The cellular nuclei of the axonal fibers were regularly aligned in the optic nerve of the control rats. In contrast, in animals with EAE, the nuclei in the nerves were grouped [26]. 

Our treatment response to this model is the prevention of demyelination by activating the ERK pathway to give rise to the expression of some anti-apoptotic proteins. However, future efforts are needed to characterize this pathway with a broader protein profile and a marker that allows us to confirm the expression of the ERK pathway in astrocytes and oligodendrocytes of the spinal cord and optic nerve.

## 4. Materials and Methods

All experiments were conducted using male Sprague Dawley rats weighing 250–300 g and of 60 days postnatal age. The rats were housed under controlled environmental conditions with a constant temperature of 23 ± 2 °C and a 12:00 h light–dark cycle. They had access to food and water ad libitum.

### 4.1. EAE Model

The induction of the EAE model was carried out according to the following technique. A total of 150 μL of the immunization emulsion was administered subcutaneously and prepared with a 1:1 homogenate of the spinal cord and pig brain, which were sources of neuroantigens, and resuspended in a proportion 1:2 Freund’s complete adjuvant. The biological material (pig spinal cord and brain) was obtained from healthy animals [27]. On experimental day 0, each rat was immunized with the immunization emulsion described above. The animals were monitored and evaluated daily throughout the experimental period. The study model of this research work is the type of EAE that resembles relapsing–remitting multiple sclerosis (Figure 5).

### 4.2. Evaluation of the Evolution of EAE

The clinical signs have already been reported in the literature [27,28] and have been classified, with the following scale, as valid: grade 0, no symptoms; grade 1, complete tail paralysis; grade 2, mild paresis of hind limbs; grade 3, complete paralysis of one hind limb; grade 4, bilateral hind limb paralysis; and grade 5, complete paralysis (tetraplegia), moribund state, or death [27,28,29]. This model has already been validated and is considered successful when the animals reach grade 2 on the clinical scale during the relapsing–remitting cycle. These clinical measurements were performed daily during the experimental process.

### 4.3. Animals and Experimental Design

The rats were randomly divided into seven groups of 5 rats each: (1) control/untreated rats, (2) immunized/untreated rats, (3) EPO (5000 IU/Kg, s.c) treatment rats/without immunization, (4) U0126 (5 mg/kg, i.p) treatment rats/without immunization, (5) immunized/EPO treatment rats, (6) immunized/ U0126 (5 mg/Kg, i.p) treatment rats and (7) immunized/EPO and U0126 treatment rats (the U0126 was administered 30 min before each administration of EPO).

### 4.4. Obtain Optic Nerve and Spinal Cord

After the treatment periods, animals were administered a sedative, Dexmedetomidine, at a dose of 50 mg/Kg, followed by a general anesthetic, ketamine, at a dose of 80 mg/Kg and sacrificed by perfusion using 4% paraformaldehyde. Afterward, each group’s brains, spinal cord, and optic nerve were paraffin-embedded, and 4 μm thick sections were cut.

### 4.5. Automated Immunohistochemistry

The samples for IHQ were processed automatically using the BOND MX equipment from Leica Biosystem. The samples were mounted on electrocharged slides and then deparaffinized in an oven at 60 °C for 30 min. Subsequently, the sections were subjected to a clearing and dehydration process as a preparatory procedure to perform immunohistochemistry. The BOND Polymer Refine Detection Kit (Leica Biosystems, Deer Park, IL, USA; REF DS9800) was utilized for immunodetection, according to the manufacturer’s instructions in automated immunodetection equipment. The antigen was recovered using EDTA for 20 min. Sections were incubated for 30 min with an anti-ERK antibody (Abcam, Cambridge, CB2 0AX, UK; REF ab214036, 1:250). Immunoreactivity was revealed using 3,3′-diaminobenzidine (DAB). The tissue sections were digitized (Aperio LV1 real-time pathology system, Leica Biosystems, Deer Park, IL, USA) and subsequently analyzed using the ImageJ 8 software.

### 4.6. Luxol Fast Blue Staining

For Luxol Fast Blue Staining, the Abcam Myelin Stain kit (Abcam, Cambridge, CB2 0AX, UK; REF ab150675) was used. This staining was performed on 4 μm thick tissues of the spinal cord and optic nerve. Longitudinal and transverse sections were cut from the paraffin-embedded tissues and incubated in LFB for 2 h at 60 °C in an incubation oven. Differentiation of the section was carried out using lithium carbonate solution and Cresyl violet. LFB-stained tissue sections were examined by light microscopy, analyzed to determine whether myelin was preserved, and then digitized for later evaluation.

### 4.7. Quantification of Histological Parameters

The cellular quantification procedure with H&E in the optic nerve includes reviewing five fields at a magnification of 40×, and in the spinal cord, six fields, each comprising an area of 160 × 240 μm at a magnification of 40× (n = 4/group). In these fields, viable cells (intact membrane, homogeneity in nuclear and cytoplasmic coloring, and oval or circular shape) were recorded. Cell counting was performed manually.

The relative intensity in the tissues with LFB staining was analyzed using the Image J software, evaluating the homogeneity and the intensity of the coloration present in the tissues. For the optic nerve, five fields were assessed at a magnification of 20×, and for the spinal cord, six fields at a magnification of 20× (n = 4/group). 

The quantification of ERK-positive cells in IHC was carried out by counting the number of positive cells per field. Five fields were analyzed at a magnification of 40× in the optic nerve, and six fields were analyzed at 40× in the spinal cord. Nuclear and weak cytoplasmic staining on neuronal cells was shown in brown. The cells were manually quantified from the digitalized images to analyze H&E staining, IHC, and LFB intensity. 

### 4.8. Statistical Analysis

Statistical analysis was performed using GraphPad Prism10 software. Data were expressed as the means ± SD. Based on the results obtained from the cell coin in H&E, the intensity of LFB expression in the tissues, and the count of ERK-positive cells, a multiple comparison one-way analysis of variance and Tukey’s post hoc tests were used for statistical analysis. *p*-values ≤ 0.05 were considered statistically significant.

## 5. Conclusions

The findings of this study suggest that erythropoietin treatment reduces cellular death in EAE-induced rats by regulating the ERK pathway. In the optic nerve of Sprague Dawley rats with EAE treated with EPO, the ERK 1/2 pathway was activated in neuronal cells, activating a neuroprotective process involved in possible remyelinating processes. Further studies are needed to improve the characterization of the inflammatory profile of the EAE model treated with EPO.

## Figures and Tables

**Figure 1 ijms-25-09476-f001:**
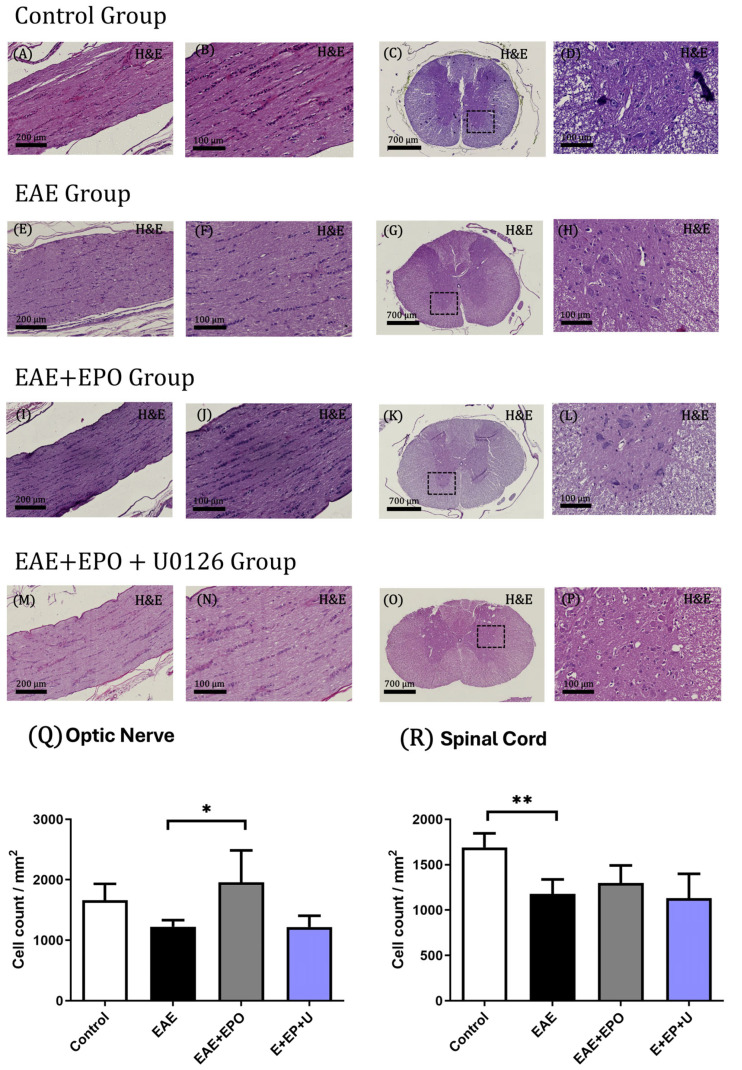
Histology of the control’s optic nerve and spinal cord tissues, EAE, EAE+EPO, and EAE+EPO+U0126 groups. These tissues were stained using the hematoxylin and eosin technique. (**A**) Optic nerve of the control group with 10× magnification; (**B**) optic nerve of the control group with 40× magnification; (**C**) spinal cord of the control group with 3× magnification; (**D**) spinal cord of the control group with 20× magnification; (**E**) optic nerve of the EAE group with 10× magnification; (**F**) optic nerve of the EAE group with 40× magnification; (**G**) spinal cord of the EAE group with 3× magnification; (**H**) spinal cord of the control group with 20× magnification; (**I**) optic nerve of the EAE+EPO group with 10× magnification; (**J**) optic nerve of the EAE+EPO group with 40× magnification; (**K**) spinal cord of the EAE+EPO group with 3× magnification; (**L**) spinal cord of the EAE+EPO group with 20× magnification; (**M**) optic nerve of the EAE+EPO+U0126 group with 10× magnification; (**N**) optic nerve of the EAE+EPO+U0126 group with 40× magnification; (**O**) spinal cord of the EAE+EPO+U0126 group with 3× magnification; (**P**) spinal cord of the EAE+EPO+U0126 group with 20× magnification; (**Q**) graph of the cell count in the optic nerve of the study groups; (**R**) graph of the cell count in the spinal cord of the study groups. The data are presented as mean ± SD obtained for the four animals per group. A one-way ANOVA and a Tukey post hoc test were performed as a statistical test. The significant differences are * *p* < 0.053 EAE vs. EAE+EPO for the optic nerve and ** *p* < 0.016 control vs. EAE in the spinal cord.

**Figure 2 ijms-25-09476-f002:**
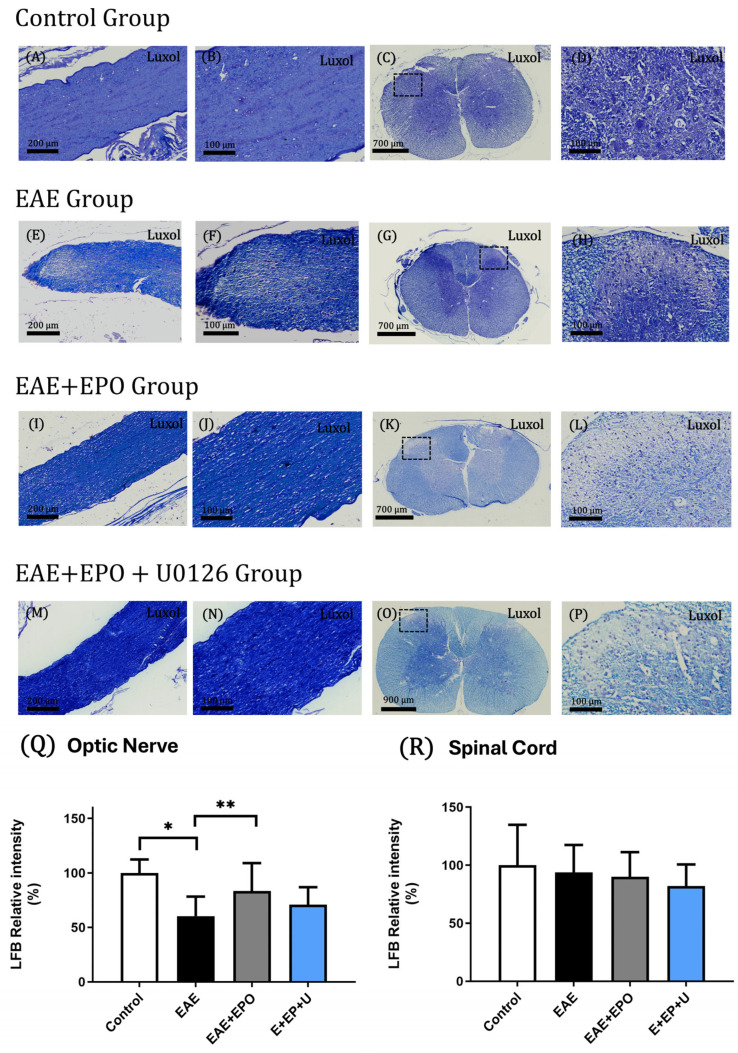
Histology of the control’s optic nerve and spinal cord tissues, EAE, EAE+EPO, and EAE+EPO+U0126 groups. These tissues were stained using the Luxol Fast Blue technique. (**A**) Optic nerve of the control group with 10× magnification; (**B**) optic nerve of the control group with 20× magnification; (**C**) spinal cord of the control group with a 2.5× magnification; (**D**) spinal cord of the control group with 20× magnification; (**E**) optic nerve of the EAE group with 10× magnification; (**F**) optic nerve of the EAE group with 20× magnification; (**G**) spinal cord of the EAE group with 2.5× magnification; (**H**) spinal cord of the control group with 20× magnification; (**I**) optic nerve of the EAE+EPO group with 10× magnification; (**J**) optic nerve of the EAE+EPO group with 20× magnification; (**K**) spinal cord of the EAE+EPO group with 2.5× magnification; (**L**) spinal cord of the EAE+EPO group with 20× magnification; (**M**) optic nerve of the EAE+EPO+U0126 group with 10× magnification; (**N**) optic nerve of the EAE+EPO+U0126 group with 20× magnification; (**O**) spinal cord of the EAE+EPO+U0126 group with 2.5× magnification; (**P**) spinal cord of the EAE+EPO+U0126 group with 20× magnification; (**Q**) graph of the LFB relative intensity in the optic nerve of the study groups; (**R**) graph of the LFB relative intensity in the spinal cord of the study groups. The data are presented as mean ± SD obtained for the four animals per group. A one-way ANOVA and a Tukey post hoc test were performed as a statistical test. The significant differences are * *p* < 0.00007 control vs. EAE and ** *p* < 0.003 EAE vs. EAE+EPO for optic nerve.

**Figure 3 ijms-25-09476-f003:**
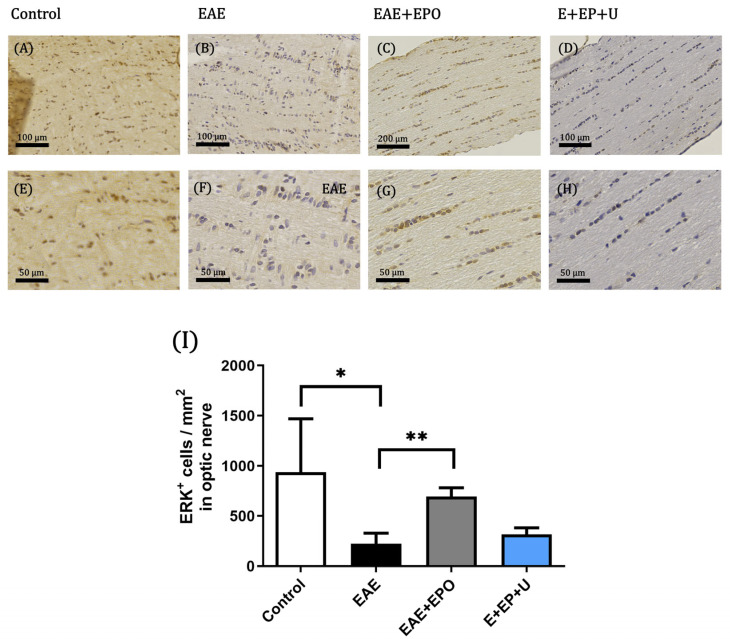
Immunohistochemistry of the ERK protein. Positive expression of ERK^+^ protein of cells in the optic nerve tissue; (**A**) positive ERK expression in the optic nerve of the control group at 10× magnification; (**E**) ERK^+^ expression in the optic nerve of the control group at 40× magnification; (**B**) positive ERK expression in the optic nerve of the EAE group at 10× magnification; (**F**) ERK^+^ expression in the optic nerve of the EAE group at 40× magnification; (**C**) positive ERK expression in the optic nerve of the EAE+EPO group at 10× magnification; (**G**) ERK^+^ expression in the optic nerve of the EAE+EPO group at 40× magnification; (**D**) positive ERK expression in the optic nerve of the EAE+EPO+U0126 group at 10× magnification; (**H**) ERK^+^ expression in the optic nerve of the EAE+EPO+U0126 group at 40× magnification; (**I**) graph of the ERK^+^ expression in neuronal cells in the optic nerve of the study groups. The data are presented as mean ± SD for the four animals per group. A one-way ANOVA and a Tukey post hoc test were performed as a statistical test. The significant differences are * *p* < 0.0013 control vs. EAE and ** *p* < 0.014 EAE vs. EAE+EPO for optic nerve.

**Figure 4 ijms-25-09476-f004:**
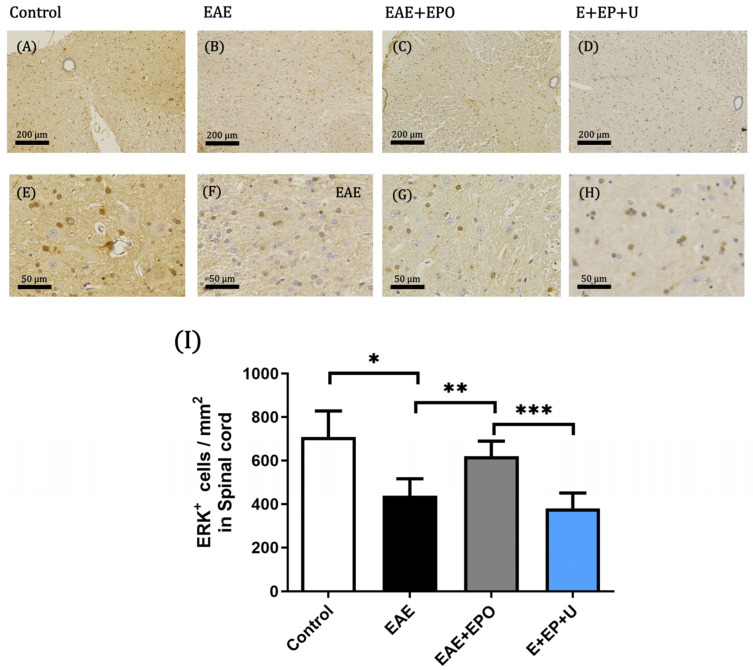
Immunohistochemistry of the ERK protein. Positive expression of ERK^+^ protein of neuronal cells in the spinal cord tissue; (**A**) positive ERK expression in the spinal cord of the control group at 10× magnification; (**E**) ERK^+^ expression in the spinal cord of the control group at 40× magnification; (**B**) positive ERK expression in the spinal cord of the EAE group at 10× magnification; (**F**) ERK^+^ expression in the spinal cord of the EAE group at 40× magnification; (**C**) positive ERK expression in the spinal cord of the EAE+EPO group at 10× magnification; (**G**) ERK^+^ expression in the spinal cord of the EAE+EPO group at 40× magnification; (**D**) positive ERK expression in the spinal cord of the EAE+EPO+U0126 group at 10× magnification; (**H**) ERK^+^ expression in the spinal cord of the EAE+EPO+U0126 group at 40× magnification; (**I**) graph of the ERK^+^ expression in neuronal cells in the spinal cord of the study groups. The data are presented as mean ± SD for the four animals per group. A one-way ANOVA and a Tukey post hoc test were performed as a statistical test. The significant differences are * *p* < 0.0008 control vs. EAE, ** *p* < 0.021 EAE vs. EAE+EPO, and *** *p* < 0.0026 EAE+EPO vs. E+EP+U for spinal cord.

**Figure 5 ijms-25-09476-f005:**
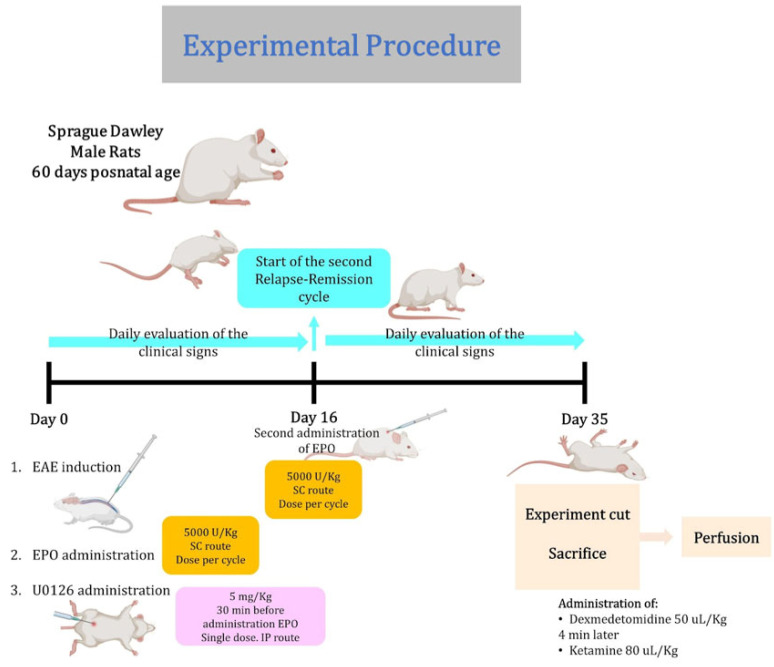
Scheme of the experimental procedure.

## Data Availability

Data are contained within the article.

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
