# Peer review of "The Neuroprotective Effect of Erythropoietin on the Optic Nerve and Spinal Cord in Rats with Experimental Autoimmune Encephalomyelitis through the Activation of the Extracellular Signal-Regulated Kinase 1/2 Signaling Pathway"

_ijms, 2024, doi:10.3390/ijms25179476_

Round 1

Reviewer 1 Report

Comments and Suggestions for Authors

The study is based on a clever idea The proposed hypothesis is valid and the method well designed. On the other hand, micro-photographs do not support the presented results and, in general, seem like selected with not much effort. A more consistent approach to the presentation of the histology is needed. Furthermore, the method of quantification of the cellular and histological parameters is not thoroughly explained, which in combination with the luck of thoroughness in the presented data raise questions about the sanity of the results.

Suggestions:

-       Explain thoroughly the methods of quantification.

-       Select better, clearer, suitably oriented, more representative micro-photographs to support your results.

Detailed Comments/ Suggestions:

Replace the word “slice” with the most accurate histologically “section”.

page 1, lines 20-21: The purpose of administering ERK1/2 is not mentioned here, and this is confusing. I suggest a brief explanation.

page 4, line 89: "With LFB staining in EAE animals, the optic nerve, and the spinal cord reveal signs"

Figure 2:

-       It would be better for the reader if tissues in the micro-photographs had similar orientation.

-       Furthermore, in the control group you focus on the ventral horn whereas in all other groups you focus on the dorsal horn. Why?

-       The staining intensity in the different tissue blocks varies greatly and this is very disorientating.

-       Bars on the micro-photographs are non-visible.

-       In general, the micro-photographs are inadequate for comparisons among the different groups.

-       lines 119-120: I genuinely can't understand how these results arrive from micro-photographs like the here presented and how the graphs correspond to the reported results.

-       How do you explain the values of LFB relative intensity of the optic nerve of the E+EP+U group?

page 6, line 126: " ERK1/2 expression"

Figure 3:

-       The selection of the presented micro-photographs does not correspond at all with the commented results. Especially the 75% loss of ERK1/2 expression in the EAE group is not shown.

-       The higher magnification micro-photographs do not represent the highlighted area (on the lower magnification) and in some cases have changed orientation. Why? This is very disorientating.

Figure 5, line 228: Do you mean "estimated"?

page 10, lines 248-250: I suggest the following: "animals were administered a sedative, Dexmedetomidine, at a dose of 50 mg/Kg, followed by a general anesthetic, Ketamine, at a dose of 80 mg/Kg and sacrificed by perfusion of 4% paraformaldehyde."

page 10, line 253: This is very confusing. Five animals of each group (that means all animals) where processed for Immunohistochemistry with pentobarbital. Which animals were sedated with ketamine?

page 10, line 256: Thickness of coronal sections?

page 12, reference 29: Although I can read Spanish, I think that it would be better for the broader audience if you could refer to an English version of the article or to a relative article in English.

Author Response

We appreciate the time you took to review our work and the valuable comments and suggestions. You will see the changes highlighted in yellow in the new file, and the entire manuscript was checked for English grammar and spelling.

Comments 1: Explain thoroughly the methods of quantification.

Response 1: Thank you for pointing this out. A methodological section explaining the quantification method was added on page 11, lines 298-318.

Comments 2: Select better, clearer, suitably oriented, more representative micro-photographs to support your results.

Response 2: The micro-photographs were replaced, and others were visually enhanced.

Comments 3: Replace the word “slice” with the most accurate histologically “section”.

Response 3: the words were replaced.

Comments 4: page 1, lines 20-21: The purpose of administering ERK1/2 is not mentioned here, and this is confusing. I suggest a brief explanation.

Response 4: ERK 1/2 was not administered; instead, a blocker of this signaling pathway was used. The paragraph was rewritten for better understanding.

Comments 5: page 4, line 89: "With LFB staining in EAE animals, the optic nerve, and the spinal cord reveal signs"

Response 5: Rewritten for better understanding. Page 4, lines 94-96.

Comments 6: It would be better for the reader if tissues in the micro-photographs had similar orientation.

Response 6: Images have been changed to the correct orientation.

Comments 7: Furthermore, in the control group you focus on the ventral horn whereas in all other groups you focus on the dorsal horn. Why?

Response 7: It was a focus error. The orientation of the image was corrected and indicated correctly.

Comments 8:  The staining intensity in the different tissue blocks varies greatly and this is very disorientating.

Response 8: The images were replaced with others that have similar intensities.

Comments 9:  Bars on the micro-photographs are non-visible.

Response 9: The bars are placed in black.

Comments 10:  In general, the micro-photographs are inadequate for comparisons among the different groups.

Response 10: The images are changed to make them more representative.

Comments 11: lines 119-120: I genuinely can't understand how these results arrive from micro-photographs like the here presented and how the graphs correspond to the reported results.

Response 11: A methodological section was added explaining the quantification method.

Comments 12: How do you explain the values of LFB relative intensity of the optic nerve of the E+EP+U group?

Response 12:  Upon reviewing this point and reanalyzing the data, we realized there was an error in graphing it, so we replaced it with the correct graph.

Comments 13: page 6, line 126: " ERK1/2 expression"

Response 13: It was corrected.

Comments 14:   The selection of the presented micro-photographs does not correspond at all with the commented results. Especially the 75% loss of ERK1/2 expression in the EAE group is not shown.

Response 14: The images were changed to more representative ones.

Comments 15: The higher magnification micro-photographs do not represent the highlighted area (on the lower magnification) and, in some cases, have changed orientation. Why? This is very disorientating.

Response 15: The images were changed.

Comments 16: Figure 5, line 228: Do you mean "estimated"?

Response 16: The image of the methodology was corrected.

Comments 17: page 10, lines 248-250: I suggest the following: "animals were administered a sedative, Dexmedetomidine, at a dose of 50 mg/Kg, followed by a general anesthetic, Ketamine, at a dose of 80 mg/Kg and sacrificed by perfusion of 4% paraformaldehyde."

Response 17: The paragraph was changed to the suggested one.

Comments 18: page 10, line 253: This is very confusing. Five animals of each group (that means all animals) where processed for Immunohistochemistry with pentobarbital. Which animals were sedated with ketamine?

Response 18: Text corrected. All animals in this experiment were sedated with dexmedetomidine and ketamine.

Comments 19: page 10, line 256: Thickness of coronal sections?

Response 19: The text was modified because we used an automated immunohistochemistry technique for these experiments, leading to variations in the methodology.

Comments 20: page 12, reference 29: Although I can read Spanish, I think that it would be better for the broader audience if you could refer to an English version of the article or to a relative article in English.

Response 20: The reference was changed to the English version.

Reviewer 2 Report

Comments and Suggestions for Authors

The paper entitled "The neuroprotective effect of EPO on the optic nerve and spinal cord in rats with EAE through the activation of the ERK1/2 signaling pathway." is very interesting.

The manuscript is structured, but needs some improvements.

1) abstract change the number (1), (2), (3), and (4) in 1), 2),3), and 4). In lines 20, 23, 25, and 27, change ; in .

2) Check that all abbreviations are written in full the first time they appear in the text. See, for example, line 37 TNF-α and NFκB.

3) In the Methods the authors need to improve the desciption of imageJ type of analysis used for the quantization of immunohistochemistry.

4) About Luxol Fast Blue Staining, the authors show grafics of quantization in Figure 2, but do not explain with which method they performed the quantization, Please explain in the text this aspect.

5) the software used for the statistical anlysis is very old, the version reported in the text is "GraphPad Prism5 software", while the actual version is GraphPad Prism10.3, please check this data and the correctness of the statisical analysis.

Comments on the Quality of English Language

Minor editing of English language required

Author Response

We appreciate the time you spent reviewing our work and welcome your valuable comments and suggestions. The changes have been highlighted in yellow in the new file, and the entire manuscript was checked for English grammar and spelling.

Comments 1: abstract change the number (1), (2), (3), and (4) in 1), 2),3), and 4). In lines 20, 23, 25, and 27, change ; in .

Response 1: The recommended changes were made

Comments 2: Check that all abbreviations are written in full the first time they appear in the text. See, for example, line 37 TNF-α and NFκB.

Response 2: The complete text was revised, and the full names of the abbreviations were added.

Comments 3: In the Methods the authors need to improve the description of imageJ type of analysis used for the quantization of immunohistochemistry.

Response 3: A methodological section was added explaining the quantification method on page 11, lines 298-318

Comments 4: About Luxol Fast Blue Staining, the authors show graphics of quantization in Figure 2, but do not explain with which method they performed the quantization, please explain in the text this aspect.

Response 4: It was added in the statistical analysis section, and the statistical test used for each analysis is added in each figure caption

Comments 5: the software used for the statistical analysis is very old, the version reported in the text is "GraphPad Prism5 software", while the actual version is GraphPad Prism10.3, please check this data and the correctness of the statistical analysis.

Response 5: The statistical software version was updated; when the analysis was run again, no variation was found in the data.

Round 2

Reviewer 2 Report

Comments and Suggestions for Authors

The authors performed the modification suggested. Now, the paper can be accepted in this form.